Diffusion models in text generation: a survey

Yi Qiuhua 1
Chen Xiangfan 1
Zhang Chenwei 2
Zhou Zehai 1
Zhu Linan 1
http://orcid.org/0000-0003-2698-3319 Kong Xiangjie 1 xjkong@ieee.org
1 College of Computer Science and Technology, Zhejiang University of Technology , HangZhou , China
2 School of Faculty of Education, University of Hong Kong , Hong Kong , China
Zubiaga Arkaitz
Electronic publication date: 2024 Feb 23
Publication date: 2024
Volume: 10
Electronic Location ID: e1905
Received 2023 Sep 28; Accepted 2024 Jan 31
Copyright: © 2024 Yi et al.
Copyright year: 2024
Copyright holder: Yi et al.
License: This is an open access article distributed under the terms of the Creative Commons Attribution License, which permits unrestricted use, distribution, reproduction and adaptation in any medium and for any purpose provided that it is properly attributed. For attribution, the original author(s), title, publication source (PeerJ Computer Science) and either DOI or URL of the article must be cited.
License URL: https://creativecommons.org/licenses/by/4.0/

Keywords: Diffusion models, Text generation, Natural language generation

Funding: National Natural Science Foundation of China 62072409, 62176234 This work was supported by the National Natural Science Foundation of China (No. 62072409, 62176234). The funders had no role in study design, data collection and analysis, decision to publish, or preparation of the manuscript.

==============================
Diffusion models are a kind of math-based model that were first applied to image generation. Recently, they have drawn wide interest in natural language generation (NLG), a sub-field of natural language processing (NLP), due to their capability to generate varied and high-quality text outputs. In this article, we conduct a comprehensive survey on the application of diffusion models in text generation. We divide text generation into three parts (conditional, unconstrained, and multi-mode text generation, respectively) and provide a detailed introduction. In addition, considering that autoregressive-based pre-training models (PLMs) have recently dominated text generation, we conduct a detailed comparison between diffusion models and PLMs in multiple dimensions, highlighting their respective advantages and limitations. We believe that integrating PLMs into diffusion is a valuable research avenue. We also discuss current challenges faced by diffusion models in text generation and propose potential future research directions, such as improving sampling speed to address scalability issues and exploring multi-modal text generation. By providing a comprehensive analysis and outlook, this survey will serve as a valuable reference for researchers and practitioners interested in utilizing diffusion models for text generation tasks.

Introduction

Diffusion-based generation

With the development of artificial intelligence, people are no longer satisfied with merely classifying data and have begun to explore how to generate new data. Currently, the most popular deep learning generative models include variational autoencoders (VAE) (Kingma & Welling, 2013), generative adversarial networks (GANs) (Goodfellow et al., 2014), flow-based generative models (Dinh, Krueger & Bengio, 2014), and diffusion models that has been widely used in the past 2 years. The essence of deep generative models is to generate new data samples that are as similar as possible to the distribution of the given training data (Harshvardhan et al., 2020). Of the three aforementioned model types, VAE must choose a variational posterior distribution, GAN requires training an additional discriminator, and the flow-based generative model requires the model to be an invertible function. Does there exist a deep generative model that only needs to train a generator without additional training of other networks or other such restrictions? The diffusion model provides one answer.

Diffusion models can be traced back to 2015, when Sohl-Dickstein et al. (2015) proposed the concept of diffusion probabilistic models (DPM). However, these models were not extensively developed during the next few years. In work published in 2020, Google improved the details of the model, introduced denoising diffusion probabilistic models (DDPM) (Ho, Jain & Abbeel, 2020) and applied them to the field of image generation, gradually bringing diffusion models into focus. After the release of DDPM, denoising diffusion implicit models (DDIM) (Song, Meng & Ermon, 2020) further improved the denoising process of DDPM, laying the foundation for subsequent diffusion models. After that, ablated diffusion model (ADM) (Dhariwal & Nichol, 2021) achieved the first victory over generative adversarial networks (GANs), causing a surge of interest in the diffusion model field. Building upon the methods of conditional image generation (Liu et al., 2021; Ho & Salimans, 2022), Palette (Saharia et al., 2022) demonstrated the immense potential of diffusion models in image-to-image translation. Additionally, GLIDE (Nichol et al., 2021), DALL ⋅E 2 (Ramesh et al., 2022), and Imagen (Saharia et al., 2022) have achieved new state-of-the-art results in the field of text-to-image generation. Later on, researchers proposed the use of diffusion models for audio generation (Kong et al., 2021; Chen et al., 2020; Kameoka et al., 2020) and achieved tremendous success.

There is no doubt that diffusion models have proved highly successful in generating content in continuous spaces, particularly in the domains of images and audio (Yang et al., 2022). Models represented by Stable Diffusion (Rombach et al., 2022) and AudioLDM (Liu et al., 2023) are diffusion models in the continuous domain, both based on latent diffusion models (LDMs), which introduce random noise to latent variables and reverse this process through a series of denoising steps to learn data generation. But how can they be applied to text generation tasks? One of the most direct challenges of applying diffusion models to the field of natural language processing (NLP) is the difference in data structure. Images exist in a continuous space, while text is discrete. To address this issue, there are two solution: one is to map the discrete text to a continuous space (Li et al., 2022b; Gong et al., 2022; Yuan et al., 2022; Strudel et al., 2022), specifically by using an embedding layer to map the text into a continuous representation space. Another approach is to preserve the discrete nature of the text and generalize the diffusion models to handle discrete data (Reid, Hellendoorn & Neubig, 2022; He et al., 2023). These two ways of applying diffusion models to NLP have achieved many excellent results in the last 2 years, with Fig. 1 illustrating the development of text generation diffusion models along the time.

Figure 1 The development of text generation diffusion models.

Scope of this survey

Due to the increasing number of publications on diffusion text generation models (see Fig. 2), it is essential to conduct a comprehensive review to summarize recent research methods and forecast future research directions. In 2023, scholars began to attempt to summarize the application of diffusion models in NLP. Zhu & Zhao (2023) provide an overview of the application of diffusion models in NLP. While the review discusses the use of diffusion models in text generation, text-driven image generation, text-to-speech, etc., it fails to provide inspiring guidance for potential future research directions. Li et al. (2023) and Čeović et al. (2023) review recent advances of diffusion models in NAR (non-autoregressive) text generation and discuss optimization techniques for text diffusion models. Zou, Kim & Kang (2023) summarize diffusion model methods in NLP and provide a comprehensive comparison with other text generation approaches. Although Li et al. (2023), Čeović et al. (2023) and Zou, Kim & Kang (2023) all offer a comprehensive summary of algorithms for diffusion models in NLP, it is unfortunate that they all focus on the perspective of applying diffusion models to the textual domain, specifically dividing them into discrete text diffusion models and continuous text diffusion models.

Figure 2 The current number of articles on diffusion text generation models.

In our work, distinguished from previous related reviews, we introduce a completely new classification perspective to categorize and summarize the research on the application of diffusion models to text generation tasks. The main contributions of this article are as follows: Provide a comprehensive overview of the latest advances in relevant current research and help researchers develop a deeper understanding of diffusion language models.

Classify studies from the novel perspective of text generation tasks and provide detailed descriptions of the methods.

Differentiate diffusion models from pre-trained language models from various perspectives, providing readers with insightful comparisons.

Elaborate on the existing challenges and expected future research directions, providing insights for researchers in relevant fields.

Survey methodology

Regarding the topic of “diffusion models in text generation”, we carried out extensive study on research questions, searched and organized the relevant literature. The research methodology primarily outlines data sources, search strategy, and literature inclusion criteria.

Research questions

Our literature review aims to address the following research questions (RQs): RQ1: How diffusion models evolve and develop?

RQ2: How are diffusion models applied to various text generation tasks?

RQ3: What are the differences between text diffusion models and pre-trained language models?

RQ4: What are the potential research directions for text diffusion models?

where the first two questions aim to illustrate the application of diffusion models in text generation, the third question is used to compare text diffusion models with pre-trained models, and the last one is intended to assist researchers in proposing potential directions for improving text diffusion models.

Data sources and research strategy

We utilized search engines such as Google Scholar, IEEE Xplore, WoS, Arxiv, and others to search and collect relevant literature. The keywords used for literature search included “diffusion model”, “text generation”, “NLP”, “pre-trained language model”, etc. Table 1 presents the data sources, search string and links.

Table 1 The description of data sources, search string and links.

Search engine	Search string	Links	
Google Scholar	Diffusion model AND text generation	https://scholar.google.com/	
IEEE Xplore	Text generation OR pre-trained language model	https://ieeexplore.ieee.org/	
WoS	Diffusion model OR text generation	https://www.webofknowledge.com/	
Arxiv	Diffusion model AND NLP	https://arxiv.org/	

Criteria for inclusion/exclusion

After searching for relevant literature, our inclusion criteria for the research are that the articles must be written in English and should be research articles. In addition, we filtered out research articles that focused on applications of diffusion models in domains other than text, such as visual and audio. Finally, we summarized the number of articles, as shown in Fig. 2, indicating that diffusion models in text generation are still in development with significant growth potential.

Definitions

Natural language generation (NLG)

Natural text generation aims to produce fluent, reasonable and understandable linguistic text from input data (Yu et al., 2022b). This task is more formally known as “natural language generation” in the literature. At present, it is one of the most important and challenging subtasks in NLP.

NLG has two principal generative methods: autoregressive (AR) and non-autoregressive (NAR), also known as end-to-end generation. With the rise of deep learning in recent years, researchers have proposed various models to realize language generation, including the Transformer (Vaswani et al., 2017), BERT (Devlin et al., 2018), and GPT (Radford et al., 2019), as well as diffusion-based text generative model. In the era of Large Language Models (LLMs), decoder-only models, exemplified by the GPT, have emerged as a pivotal technology in the domain of text generation. Such models generate text exclusively through the decoder, obviating the necessity for a dedicated encoder, and operate in an autoregressive manner, sequentially generating discrete tokens. The introduction of diffusion-based models has steered the evolution of the text generation field towards harnessing both discrete and continuous features more comprehensively across diverse tasks.

Text generation tasks

To date, researchers have developed many techniques with regard to text generation applications (Li et al., 2021a). NLG encompasses many downstream subtasks that take various forms of data as input (Celikyilmaz, Clark & Gao, 2020). Examples include unstructured inputs, such as sentences and paragraphs; structured inputs, such as graphs and tables; and multimedia inputs, such as images, videos, and speech (Li et al., 2021b). Figure 3 illustrates the typical text generation task.

Figure 3 Subtasks for text generation.

Diffusion model

Diffusion models (Sohl-Dickstein et al., 2015; Ho, Jain & Abbeel, 2020) were originally latent variable models designed for continuous data domains. The model training process can be divided into two steps: the forward noise addition process and the reverse denoising process.

The forward process originates from data x0∼q(x). The model adds the noise corresponding to time step t and obtains output xt according to xt−1. At step T (the final time step) to obtain xT, the data is transformed into an invisible noise distribution. In the reverse process, according to the given condition xt ( t decrements from T to 0), the Bayes’ theorem is used to determine xt−1. As a result, the target sentence or image can be generated by iteratively sampling noise.

Specifically, given an initial sample x0, a small amount of Gaussian noise is gradually injected into the sample according to the forward process q(xt|xt−1) during each step to disrupt the original data. q(xt|xt−1) is represented by the following equation:

(1) q(xt|xt−1)=N(xt;1−βtxt−1,βtI)

where βt=1−αt is a pre-defined noise schedule (Li et al., 2023). The noise added at each step is independent and follows a normal distribution. As the number of iterations increases, the intensity of the added noise also increases, requiring the intermediate latent variables to incorporate more noise to effectively disrupt the training data. Consequently, βt will progressively increase over time, eventually transforming x0 into random noise, approximately following a normal distribution N(0,I).

Each iteration of the forward process will produce a new latent variable xt. Therefore, the diffusion model can model the original data x0 as a Markov chain x0,x1,x2,⋯,xT. Based on the re-parameterization method, the model can convert q(xt∣xt−1) into q(xt∣x0) for better sampling:

(2) q(xt∣x0)=N(xt;α¯tx0,1−α¯tI)

where α¯ = σi=1tαi. Since the reverse denoising continuously approaches the posterior distribution q(xt−1∣xt), the denoising model utilizes pθ(xt−1∣xt) to restore xT to the desired result. The denoising process can be formulated as follows:

(3) pθ(xt−1∣xt)=N(xt−1;μθ(xt,t),∑θ(xt,t))

where μθ(xt,t) and ∑θ(xt,t) can be computed using U-Net or the Transformer model (Li et al., 2022b). The variance ∑θ(xt,t) is determined by a specific scheduler and remains fixed, hence there is no need to predict it. The ultimate objective of the training process is to predict μθ(xt,t).

On the basis of known x0 and forward process q(xt|xt−1), using Bayes’ formula can directly link xt and x0 in the denoising process instead of tediously using xt to predict xt−1 step by step. As a result, the final training goal can be simplified as follows:

(4) Lsimple=∑t=1TEq[||μt(xt,x0)−μθ(xt,t)||2]

where μt is the mean of posterior q(xt−1|xt,x0). The objective of the model is to minimize the mean square error between the two distributions.

Diffusion models in text generation

In this section, we will elaborate sequentially on diffusion models in the field of text generation. We categorize these into three types based on the tasks of text generation: conditioned text generation, unconstrained text generation, and multi-mode text generation. Table 2 provides a summary and comparison of all diffusion text models considered in this survey.

Table 2 Summary of diffusion models in text generation, grouped by type.

Model	Noise schedule	Sampling	Space	Generation process	Pretrain	
Conditional text generation (Text-driven generation)	
DiffuSeq	Partial noising	Minimum Bayes Risk	Ca	NARb	/	
DiffuSum	Partial noising	/	C	NAR	/	
DiffusER	Edit-based reconstruction	Beam search, 2D Beam search, Nucleus sampling	D	NAR	/	
SeqDiffuSeq	Adaptive noise schedule	Self-conditioning	C	NAR	/	
Zero-Shot Diffusion	Partial noising	Classifier-free conditional denoising	C	NAR	/	
GENIE	/	Continuous paragraph denoise	C	NAR	Arge-scale pretrained diffusion language model	
RDMs	Mask	Reparameterized sampling, stochastic routing mechanism	D	NAR	Pre-trained autoregressive Transformer	
Diffusion-NAT	Mask	Self-prompting	D	NAR	BART	
CDCD	Time warping	Inverse transform sampling, time warping	C	NAR	BERT	
DiNoiSer	Manipulated noises	MBR	C	NAR	/	
AR-DIFFUSION	Square-root	Multi-level diffusion strategy, dynamic movement speeds, MBR	C	AR	/	
Conditional text generation (Fine-grained control generation)	
Diffusion-LM	Cosine	MBR	C	NAR	/	
Masked-Diffuse LM	Strategically soft-masking	MBR	D	NAR	BERT	
Difformer	Sqrt noise	2D parallel decoding	C	NAR	/	
Text-driven generation and Fine-grained control generation	
LDEBM	/	/	C	NAR	/	
Unconstrained text generation	
D3PM	Uniform transition matrices	/	D	NAR	/	
DiffusionBERT	Spindle schedule	x0-Parameterization	D	NAR	BERT	
Multi-mode text generation	
SED	Span masking	Self-conditioning	C	NAR	Embedding pretraining	
SUNDAE	Uniform transition matrices	Unrolled denoising, low-temperature sampling, argmax-unrolled decoding, updating fewer tokens	C	NAR	/	
LD4LG	Cosine	Self-conditioning	C	NAR	BART	
SSD-LM	Logits-generation	Sampling, multi-hot and greedy	C	NAR	/	
Notes:

a “C” and “D” respectively represent continuous and discrete.

b “AR” and “NAR” respectively stand for autoregressive and non-autoregressive.

Conditional text generation

Text-driven generation

The objective of text-driven generation is to generate a target sentence y=y1,y2,⋯,yT given a source sentence x(i)=x1(i),x2(i),⋯,xL(i), with the goal of maximizing the conditional probability P(y|x). Specifically, the objective function can be expressed as: argmaxθP(y|x;θ), where θ represents the parameters of the model, and P(y|x;θ) denotes the conditional probability of generating the target text y given the input text x. The sequence-to-sequence conditional text generation typically uses the encoder-decoder architecture (Lee, Lee & Hwang, 2020), schematically shown in Fig. 4. Currently, diffusion-based text generation approaches predominantly utilize text-driven conditional generation; the following is a detailed description of diffusion models for text-driven generation.

Figure 4 Text-driven generation.

DiffuSeq (Gong et al., 2022) is a groundbreaking conditional diffusion language model that applies diffusion to sequence-to-sequence (SEQ2SEQ) text generation tasks. Notably, DiffuSeq introduces the concept of partial noising, which selectively applies Gaussian noise to the target sequence while preserving the integrity of the source sentence embeddings. This innovative approach allows for controlled corruption and enhances the generation process.

DiffuSum (Zhang, Liu & Zhang, 2023) extends the idea of conditional diffusion modeling to the task of text summarization. Similar to DiffuSeq (Gong et al., 2022), which employs partial noise in the diffusion process, DiffuSum goes a step further by incorporating additional components, such as matching loss and multiclass contrast loss. This pioneering research on DiffuSum represents the first dedicated exploration of text summarization using diffusion models.

DiffusER (Reid, Hellendoorn & Neubig, 2022) differs from the traditional diffusion model in terms of noise injection. It considers operations such as insertion, deletion, and editing as forms of noise, because both Gaussian noise and these editing operations are in essence destroying the original data. Such an operation fully takes into account the discrete characteristics of the text, making the generation more flexible.

SeqDiffuSeq (Yuan et al., 2022), an encoder-decoder Transformers architecture, incorporates two key techniques: adaptive noise schedule and self-conditioning, resulting in substantial enhancements in both the quality and speed of text generation.

Zero-shot diffusion (Nachmani & Dovrat, 2021), inspired by encoder-decoder architecture, inputs the source language sentence x (i.e., the condition) into the Transformer encoder and the noisy target language sentence y into the decoder. Notably, this work is the first to apply the diffusion model to conditional text generation tasks.

GENIE (Lin et al., 2022) represents a significant advancement in the field of language modeling with its large-scale pre-training approach. Using the masked source sequence s as the input of the encoder and incorporating the continuous paragraph denoise training method, GENIE has demonstrated its ability to generate text that exhibits both high quality and remarkable diversity. This not only showcases the effectiveness of diffusion language models but also opens up new possibilities for various natural language processing tasks.

RDMs (reparameterized diffusion models) (Zheng et al., 2023) introduce reparameterization and a stochastic routing mechanism, leading to two significant advantages: simplified training and flexible sampling. However, currently RDMs can only generate sentences of fixed length.

Diffusion-NAT (Zhou et al., 2023) integrates discrete diffusion models (DDM) and BART into non-autoregressive (NAR) text generation, unifying the inference and denoising processes into a masked token recovery task. Diffusion-NAT focuses on conditional text generation tasks, highlighting the synergistic effect of discrete diffusion models and pre-trained language models in enhancing text generation.

CDCD (Dieleman et al., 2022) improves the training process of diffusion models by incorporating score interpolation and time warping techniques, achieving excellent performance in language modeling and machine translation tasks.

DiNoiSer (Ye et al., 2023) argues that simply mapping discrete tokens to continuous space through embedding is not sufficient to fully eliminate the discrete nature of text. Therefore, DiNoiSer employs counter-discreteness training by utilizing adaptive noise levels and amplifies the noise scale to leverage source conditions, leading to consistent improvements across multiple conditional text generation tasks.

Difformer (Gao et al., 2022), a denoising diffusion model built upon the Transformer architecture, tackles the challenges of diffusion models in continuous embedding space. By incorporating an anchor loss function, a layer normalization module for embeddings, and a noise factor for Gaussian noise, Difformer exhibits remarkable benefits in machine translation and text summarization tasks.

AR-DIFFUSION (Wu et al., 2023), unlike most text diffusion models, proposes a multi-level diffusion strategy and dynamic movement speeds to explore an autoregressive text generation diffusion model and demonstrates strong performance even with very few decoding steps.

Fine-grained control generation

Fine-grained controlled text generation accepts fine-grained control conditions (sentiment, theme, style, etc.) as input and introduces a conditional variable c, which can be used to represent control attributes (Hu & Li, 2021). The generation process diagram is shown in Fig. 5. For example, in the case of sentiment-controlled generation (Zhu et al., 2022), c represents the labels of different sentiment polarities (Li et al., 2022b). The objective of controllable text generation is to maximize the conditional probability P(x|c), which represents the probability of generating a text sequence x given a specific condition c. Currently, the research on the application of diffusion models in the context of controllable text generation is still in its preliminary exploration stage.

Figure 5 Fine-grained control generation process.

Diffusion-LM (Li et al., 2022b), a controllable language model based on continuous diffusion, has been successfully applied to six fine-grained control generation tasks. However, Diffusion-LM has much room for further optimization and improvement in terms of perplexity, decoding speed, and convergence speed.

Masked-Diffuse LM (Chen et al., 2023), inspired by linguistic features, proposes to apply strategic soft-masking to corrupt text in the forward process and iteratively denoise it through direct text prediction. Compared to Diffusion-LM (Li et al., 2022b), this model has lower training cost and better performance through five controllable text generation tasks.

Latent Diffusion Energy-Based Model (LDEBM) (Yu et al., 2022a), combining diffusion models and latent space energy-based models, uses diffusion recovery likelihood learning to address poor sampling quality and instability. It exhibits superior interpretable text modeling performance in several challenging tasks such as conditional response generation and sentiment-controllable generation.

Unconstrained text generation

Unconstrained text generation (Li et al., 2022a), also known as unconditional text generation, refers to the process where a model generates text without specific themes or length limitations based on a training corpus. Currently, diffusion models have been proposed and employed for unconstrained text generation.

D3PM (Austin et al., 2021) develops a more structured categorical corruption process by using similarity between tokens to enable gradual corruption and denoising and explores inserting (MASK) token to draw parallels to auto-regressive and mask-based generative models. As a result, D3PM achieves strong results on character-level text generation while scaling to large vocabularies on LM1B (Language Model on One Billion Words).

DiffusionBERT (He et al., 2023) creatively proposes to use BERT as its backbone to perform text generation, combining pre-training models (PLMs) with a discrete diffusion model of the absorption state of the text to address the problem of unconditional text generation with non-autoregressive models. Experiments on unconditional text generation show significant improvements in perplexity and BLEU scores over D3PM (Austin et al., 2021) and Diffusion-LM (Li et al., 2022b).

Multi-mode text generation

In addition to handling the three aforementioned text generation tasks individually, current research on diffusion models in text generation often focuses on addressing multiple tasks simultaneously.

Self-conditioned embedding diffusion (SED) (Strudel et al., 2022) proposes a continuous diffusion mechanism called self-conditioned embedding, which learns a flexible and scalable diffusion model suitable for both conditional and unconditional text generation. Notably, this study can support text padding, laying the foundation for exploring embedding space design and padding capabilities.

Step-unrolled Denoising Autoencoder (SUNDAE) (Savinov et al., 2021) introduces the training mechanism of unrolled denoising based on Autoencoders. Compared to the usual denoising approach, it requires fewer iterations to converge and demonstrates good performance in machine translation and unconditional text generation tasks. Additionally, it breaks the autoregressive limitation and can fill arbitrary blank patterns in templates, paving the way for new approaches to text editing and text repair.

Latent Diffusion for Language Generation (LD4LG) (Lovelace et al., 2022), unlike other works that transfer discrete text to continuous space by embedding, learns the process of diffusion over the latent space of pre-trained language models and extends this framework from unconditional text generation to conditional text generation.

Semi-autoregressive Simplex-based Diffusion Language Model (SSD-LM) (Han, Kumar & Tsvetkov, 2023), a semi-autoregressive diffusion language model that performs diffusion over the natural vocabulary space, enables flexible output length and modularity control through these two key designs features. On unconstrained and controlled text generation tasks, SSD-LM outperforms the autoregressive baseline model in terms of quality and diversity.

Comparison between text diffusion models and plms

Large-scale pre-trained language models (PLMs) based on transformers represented by GPT (decoder-only model), BERT (encoder-only model), and T5 (encoder-decoder model) provide a strong foundation for natural language processing tasks. Among the articles published in recent years, publications based on pre-training have occupied the mainstream position, hence this survey examines the similarities and differences between PLMs and diffuison models.

Through deep learning training on a large-scale corpus, a pre-trained model can not only learn richer and more targeted semantic information, but also understand the grammar and context of natural language, and generate coherent and logical text. PLMs have shown impressive results in many NLP domains and applications. Their training process can be divided into: (1) Pre-training: PLMs first train a general and large-scale language model on large-scale text, which contains rich contextual semantic information; (2) Fine-tuning: according to different downstream tasks, the pre-training model performs discriminative learning on labeled data.

When comparing autoregressive and diffusion models, it is imperative to balance their merits and drawbacks in terms of generation speed, diversity, and other relevant factors. This consideration facilitates a judicious selection based on specific application scenarios and task requirements. In this survey, we compare PLMs and diffusion-based text generation models across the following four dimensions, as shown in Table 3 below.

Table 3 Comparison between diffusion models and PLMs.

Dimension	PLMs	Diffusion-based models	
Generation methods	Usually autoregressive.	Usually non-autoregressive.	
Discrete text handling	One-hot encoding, distributed representation, bag-of-words representation and word embedding representation.	Discrete text diffusion and continuous text diffusion.	
Time complexity	Related to factors such as the number of layers of the model, the number of attention heads, the dimension of the hidden layer, and the size of the training data.	Usually related to the number of sampling steps and the model complexity.	
Diversity of generated results	Tending to choose words with high probabilities may result in relatively conservative and similar generated outcomes.	By introducing more randomness, the generated text tends to exhibit diversity.	

Comparison of generation methods

Pre-trained language models The PLMs based on Transformers usually adopt an autoregressive approach (Manning & Schutze, 1999) (see Fig. 6), to generate sentences via a time series forecast technology. A trained language model samples a sequence of discrete words to predict the next possible word based on previous content.

Figure 6 Autoregressive language model.

Formally, the model obtains the probability score of word xi by calculating the conditional probability P(xi|x1,x2,⋯,xi−1) (see Eq. (5)). After concatenating xi behind the original sequence (x0,⋯,xi−1) to obtain the new representation (x0,⋯,xi−1,xi), the model uses the new representation to predict the probability score of next word xi+1. In this way, the next word will continuously generate in a loop until <eos> or another constraint token is generated.

(5) p(xi,xi+1,⋯,xl∣x0,x1,⋯,xi−1)=∏i=tlp(xi|x1,x2,⋯,xi−1)

Diffusion-based models The generation method of the diffusion model in NLP is different from the traditional autoregressive method. As can be seen from Fig. 7, its training process starts with an original sentence. These models generate sentences by first constantly adding noise (usually Gaussian noise) to obtain a completely invisible noise distribution, then producing a word vector through the iterative denoising of Gaussian noise. This generative approach introduces inherent stochasticity, enhancing the diversity of the generated outcomes.

Figure 7 Diffusion-based generation process.

Discrete text handling

Pre-trained language models Because of the particularities of discrete text, putting the words into the NLG model requires special processing. This mainly includes one-hot encoding, distributed representation, bag-of-words representation and word embedding representation.

One-hot encoding uses a completely different vector to represent words, which can lead to data sparsity; distributed representation represents words based on their contextual distribution, which objectively draws on human association ability. However, there are still issues such as sparsity, and even high-frequency words can mislead calculation results. The bag-of-words representation is established in cases of unordered text and works by adding all the corresponding vectors of the word to form the final text vector representation. Word embedding uses an embedding layer to map discrete features to a continuous vector space, where each eigenvalue corresponds to a unique vector. At the same time, the embedding layer can be learned through pre-training or initialized randomly and trained together with other model parameters.

Diffusion-based models Although PLMs have proven successful in text generation, their autoregressive generation method follows the left-to-right and word-by-word pattern, which poses difficulties when taking into account flexibility and controllability. To address these limitations, some researchers have proposed using diffusion models. However, a primary challenge lies in incorporating discrete text into the model. At present, there are two mainstream methods among all diffusion-based models:

Discrete text diffusion models first refine the sentence to the token level when processing the discrete diffusion model, then map different tokens to the transfer matrix through the establishment of a category distribution function. For instance, He et al. (2023) propose an absorbing state to either keep each token unchanged or convert it to a [MASK] token with a certain probability, so as to form a transfer matrix and train the matrix to convergence, i.e., all tokens change to [MASK]. However, researchers have observed that it is possible to generate an unknown marker during the token transformation process (e.g., a tag may be damaged and randomly marked with a certain probability).

Continuous text diffusion models avoid the aforementioned instability through a simple and effective technique. Qin et al. (2022) propose relaxing the output of a discrete language model to continuous variables to help learn semantic information more accurately. Continuous text diffusion models first utilize “an embedding” technique to encode the discrete text into continuous variables with low dimensionality and rich semantics, then perform forward diffusion and reverse denoising to obtain a latent variable. Finally, the discrete text is retrieved using the rounding method to map the latent variable back to words.

Overall, based on whether the input to the diffusion model is spatially continuous or not, text generation diffusion models can be classified into discrete text diffusion models (Austin et al., 2021; Reid, Hellendoorn & Neubig, 2022; Zheng et al., 2023; He et al., 2023) and continuous text diffusion models (Li et al., 2022b; Savinov et al., 2021; Gong et al., 2022; Yuan et al., 2022; Strudel et al., 2022; Lin et al., 2022). The discrete text diffusion models perform diffusion process at the token level, with the advantage of directly handling discrete text data without the need for additional embedding operations. However, its disadvantage is that it is difficult to capture the semantic information of token context. In contrast, the continuous text diffusion model employs a more stable technique by diffusing over a continuous latent space, which can contain richer textual semantic information. Nevertheless, the challenge lies in the conversion of discrete text data into continuous latent vectors, potentially leading to information loss. Each of these approaches presents unique advantages and challenges, offering extensive and profound research directions within the field of text generation.

Time complexity

Pre-trained language models Pre-trained language models are typically pre-trained on large amounts of unlabeled text data, often as autoregressive models. During the training of an autoregressive model, the elements at each position depend on the previously generated elements. The time complexity of pre-trained language models is mainly determined by factors such as the number of layers in the model, the size of the hidden layer, the number of attention heads, and the length of the input sequence. The time complexity of a given model is approximated as O(LN2D), where L denotes the number of layers, N represents the sequence length, and D signifies the hidden layer dimension.

In the generation phase, an autoregressive model must execute sampling operations, with the generation time complexity exhibiting a linear correlation with the sequence length. The forward calculation time complexity at each position is approximately O(LDN), where L is the sequence length, D is a d-dimensional vector representing each position, and N denotes the time cost of performing forward calculations at each position.

Diffusion-based models The time complexity of diffusion models is primarily contingent upon the number of sampling steps and the computational complexity per step. Within the diffusion model, the generative process involves multiple iterations, with each iteration requiring predictions facilitated by a neural network, such as the Transformer. As a result, the time complexity of diffusion models may be relatively elevated, particularly when confronted with a substantial number of sampling steps.

Currently, there is a paucity of research focused on the time complexity of diffusion models. In order to provide a more intuitive comparison of the time complexity between Diffusion Models and PLMs, we have referenced existing works and experimental data. Taking DiffusionBERT, a diffusion model based on BERT, as an example, when both models use a step of 64, the inference time of DiffusionBERT is more than twice as slow as that of GPT, as illustrated in the Table 4. It is noted in RDMs that continuous diffusion models exhibit time complexities several orders of magnitude higher than GPT2. However, RDMs achieve a running speed approximately 10 times faster than a comparable-sized autoregressive baseline like GPT2, owing to the implementation of various optimization techniques.

Table 4 Comparison of inference time.

Method	Step	Inference times (s)	
DiffusionBERT	64	4.25	
Diffusion-LM	2,000	83.67	
GPT	64	1.55	

In summary, diffusion models generally exhibit higher time complexity because they require multiple iterations to recover text from noise. In contrast, PLMs have lower time complexity as they only need a single forward pass to predict the next word from the context. The choice of an appropriate model depends on specific application scenarios and requirements. For instance, in diffusion models, the generation process typically involves parallel generation of the entire sequence, while autoregressive models must sequentially generate elements at each position. Therefore, when generating long sentences, diffusion models might be more efficient.

Diversity of generated results

For text generation tasks, we usually use evaluation metrics such as BLEU (Papineni et al., 2002), ROUGE (Lin, 2004) and MAUVE (Darling et al., 2004) to measure the quality of the generated text. In Table 5, we summarize the results of BLEU and SacreBLEU evaluations of different models on the datasets IWSLT14 (Cettolo et al., 2014), WMT14 (Bojar et al., 2014) and WMT16 (Bojar et al., 2016). From existing studies, it is observed that the text quality generated by diffusion-based models is comparable to that of autoregressive language models, and in some cases, text generated by diffusion-based models even surpasses that of autoregressive language models.

Table 5 BLEU and SacreBLEU evaluations on IWSLT14, WMT14, and WMT16 datasets.

Models	IWSLT14 DE-EN	WMT14 EN-DE	WMT16 EN-RO	
	BLEU	SacreBLEU	BLEU	SacreBLEU	BLEU	SacreBLEU	
Transformer	32.62	33.61	26.37	26.85	32.76	32.86	
CMLM	26.41	29.41	25.94	23.22	32.13	31.26	
DiffuSeq	27.03	–	13.73	15.37	23.37	25.45	
SeqDiffuSeq	28.65	–	14.37	17.14	23.98	26.17	
Difformer	32.18	–	26.5	23.8	32.52	–	
CDCD	–	–	20	19.7	–	–	
AR-DIFFUSION	35.62	32.35	–	–	–	–	
DiNoiSER	–	31.61	–	25.88	–	32.84	

The impact of result diversity on different types of tasks varies. For generation tasks such as chatbots and story generation, the diversity of generation results can enhance interactivity and creativity and improve user experience. To assess the diversity of generated texts, GENIE (Lin et al., 2022), AR-DIFFUSION (Wu et al., 2023) and DiffusionBERT (He et al., 2023) utilize SELF-BLEU (Zhu et al., 2018) (lower scores indicate higher diversity of generated text) as an evaluation metric, while Diffusion-NAT employs Distinct-1/2 (Li et al., 2015) (higher scores indicate higher diversity of generated text) as a metric. In Fig. 8, the results of the diversity comparison for some of the models are shown. From the perspective of result diversity, diffusion models demonstrate significant advantages over pre-trained language models. For instance, as shown in Fig. 8A, the diversity of text generated by AR-DIFFUSION and GENIE is significantly higher than that of the BART (Lewis et al., 2020) model. This is because the PLMs are obtained through self-supervised learning on large-scale text data, which tend to generate more common phrases and sentences, resulting in similar generated results. However, diffusion models enhance randomness in generation through techniques such as noise injection and random sampling, thereby increasing the richness and diversity of the generated text. In summary, these experimental results collectively indicate that text generated by diffusion language models presents rich diversity while maintaining quality.

Figure 8 Diffusion-based generation process.

In general, diffusion models and PLMs both possess unique advantages and limitations in the field of text generation. In terms of text quality, both models can generate smooth, coherent, and meaningful text. However, diffusion models excel in generating diversity, capable of creating text in different styles, emotions, and themes. It is important to note that diffusion models may be more prone to generating content that deviates from common sense or logic, whereas pre-trained language models may lean towards producing repetitive or irrelevant text. Regarding generation speed, diffusion models are relatively slow, requiring multiple iterations to obtain the final result. To enhance generation speed, diffusion models can adopt various acceleration techniques, such as parallelization. Additionally, diffusion models offer the capability of pluggable controllability. In summary, they each have their strengths and weaknesses and can draw inspiration from each other to achieve a better balance between generation effectiveness and user experience.

Future directions

While diffusion models have made progress in text generation, there are still various underlying challenges, such as slow convergence and long training time. In response to these challenges, researchers have proposed a range of methods and techniques aimed at enhancing the performance of diffusion models. However, diffusion models still hold significant potential for development in the field of text generation, and much exploration remains to be undertaken. In this section, we will explore several potential research directions for diffusion models in the field of text generation.

Zero-shot tasks

A diffusion model is a probabilistic inference-based generative model, which generates new samples by modeling the probability distribution of the data and random sampling. In the face of zero-shot problems, the diffusion model can leverage the learned data distribution characteristics and prior knowledge from the training phase to generate new samples. In the field of computer vision, research has shown that diffusion models have the ability to handle zero-shot problems (Xu et al., 2023a; Wang et al., 2023). Similarly, in the field of NLP, the developers of zero-shot diffusion (Nachmani & Dovrat, 2021) found that diffusion models can address zero-shot translation problems. In the future, in controllable text generation, it will be possible to control specific attributes of generated samples to satisfy specific conditions. Furthermore, the data generated by diffusion models exhibits diversity, and using diffusion models for data augmentation can to some extent address the problem of limited data.

Multimodal diffusion models

Multimodality has become a trend and has demonstrated tremendous potential in various fields (Zhu et al., 2023). Diffusion models can already handle data from different modalities (text, image, audio, etc.), and if a unified multimodal diffusion model can be constructed, the complementarity and correlation between modalities can be explored to obtain more accurate and comprehensive information, accurately understand text, and improve the performance of tasks such as sentiment analysis, visual question answering, and image description. Currently, numerous studies have successfully implemented generative diffusion models from one modality to another. For example, researchers have made significant progress in text-to-audio (Yang et al., 2023; Huang et al., 2023b, 2023a), text-to-image (Zhang, Rao & Agrawala, 2023; Ruiz et al., 2023), and image-to-text (Fujitake, 2023). In addition to the studies of single cross-modal transitions, there is a body of research proposing multimodal mutually guided generative approaches (Huang et al., 2022; Yang, Chen & Liao, 2023; Ma et al., 2023). Huang et al. (2022), for example, employed both image and text modalities to jointly guide the generation of images, achieving a higher degree of controllability. In addition, several studies have proposed unified diffusion frameworks such as UniDiffuser (Bao et al., 2023) and Versatile Diffusion (Xu et al., 2023b). UniDiffuser not only encompasses multiple functions such as images and text co-generation and images and text rewriting, but also achieves inter-modal transformation among various modalities. As for the text generation task, the unified multimodal diffusion model can explore the complementarity and correlation between modalities, so as to obtain more accurate and comprehensive information, accurately understand text, and improve the performance of tasks such as sentiment analysis, visual question answering, and image description.

Combination with PLMs

Pre-training and fine-tuning, which are widely adopted in current research, are indispensable and crucial techniques in the field of NLP. They can capture rich semantic information and reduce the consumption of computational resources. Currently, some works have combined diffusion models with pre-trained language model BERT (Dieleman et al., 2022; Chen et al., 2023; He et al., 2023). This is mainly because pre-trained language models are trained on a large corpus of text and have language modeling capabilities, while can also speed up inference. In future work, more efficient ways of integrating diffusion models with pre-trained models can be considered, such as incorporating in-context learning, prompt learning, and other techniques.

Speeding up sampling

In diffusion models, generating samples typically requires multiple iterations of computations. Some studies, such as SED (Strudel et al., 2022), have pointed out the limitations of low sampling efficiency in diffusion models, which is indeed a drawback of diffusion models. To address this issue, in the field of computer vision, there have been a few studies that propose different efficient sampling strategies (Bond-Taylor et al., 2022; Xiao, Kreis & Vahdat, 2022; Watson et al., 2022; Vahdat, Kreis & Kautz, 2021; Zhang & Chen, 2021). These methods have demonstrated the ability to double the sampling speed in many cases. In the future, we believe that in addition to designing specialized sampling strategies, it will also be possible to draw inspiration from successful sampling strategies in computer vision and apply them to the field of NLP.

Designing embedding space

In order to use the diffusion model on continuous space, it is common to map discrete text into a continuous space using an embedding. The embedding space is learnable during the training process, and the objective of embedding is to map input data to a low-dimensional vector space by learning the representation of the data. However, during the training process, in order to minimize the loss function, the embedding may map all input data to a similar embedding space, leading to the collapse of the loss function. This will cause the model to be unable to distinguish between different samples. Therefore, it is necessary to adopt certain strategies to guide the learning of the embedding space and devise better embedding space to ensure that the original data is appropriately represented.

Conclusions

This article investigates the recent progress of text diffusion models. First, we briefly introduced text generation and its subtasks, and elaborated in detail on the formula of the diffusion models. Second, we reviewed articles applying diffusion models to tasks of conditional text generation, controlled text generation, and unconstrained text generation. Third, we made a comprehensive comparison between diffusion models and the current mainstream models (PLMs), explored their differences in multiple dimensions, and emphasized the strong advantages of diffusion models in text generation.

This survey of the diffusion model provides a comprehensive overview of the tasks of conditional and unconstrained text generation. In the meantime, we also proposed some possible challenges and future research directions for diffusion models. We hope that this survey can promote the progress of diffusion models in the NLP field.

We sincerely thank all those who provided support and assistance during the research process for the completion of this survey.

Additional Information and Declarations

Competing Interests

Author Contributions

Data Availability

Xiangjie Kong is an Academic Editor for PeerJ.

Qiuhua Yi conceived and designed the experiments, performed the experiments, analyzed the data, performed the computation work, prepared figures and/or tables, authored or reviewed drafts of the article, and approved the final draft.

Xiangfan Chen conceived and designed the experiments, performed the experiments, analyzed the data, performed the computation work, prepared figures and/or tables, authored or reviewed drafts of the article, and approved the final draft.

Chenwei Zhang conceived and designed the experiments, performed the experiments, analyzed the data, performed the computation work, authored or reviewed drafts of the article, and approved the final draft.

Zehai Zhou conceived and designed the experiments, performed the experiments, analyzed the data, performed the computation work, prepared figures and/or tables, authored or reviewed drafts of the article, and approved the final draft.

Linan Zhu conceived and designed the experiments, performed the experiments, analyzed the data, performed the computation work, authored or reviewed drafts of the article, and approved the final draft.

Xiangjie Kong conceived and designed the experiments, performed the experiments, analyzed the data, performed the computation work, authored or reviewed drafts of the article, and approved the final draft.

The following information was supplied regarding data availability:

This is a literature review.

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
