# Peer review of "Diffusion models in text generation: a survey"

_PeerJ Computer Science, doi:10.7717/peerj-cs.1905_

## Round 0.1 · original submission · Major Revisions

This is a detailed and interesting survey paper on diffusion models for text generation. While there are numerous positive aspects in the paper, there are areas where the paper could be improved, not least: (i) adding a review and discussion on decoder-only architectures, (ii) flesh out the analysis of time complexity of diffusion models, (iii) provide a stronger motivation to the need oft his survey paper as well as its differentiation from related surveys, (iv) consider whether Figure 6 is needed, as pointed out by R2, and (v) provide a table comparing results of autoregressive and diffusion models on the same datasets, to enable easier comparison.

Please provide a point-by-point response to reviewer concerns as you revise the submission.

Reviewer 1 ·

Basic reporting

The survey paper on applying diffusion models in text generation is within the scope of the journal.

I don't think this survey paper is necessary in the field. The field has been reviewed several times, as described in the 'Scope of this survey' section. This review is not too different from the existing reviews, and I don't really think a small and concentrated field (only about 30 papers, as shown in Figure 1) needs such many reviews.

The introduction is clear and easy to follow.

Experimental design

The survey methodology clearly missing the decoder-only architecture of the text generation methods, which are the mainstream right now.

The review is well-organized and includes reasonable citations.

Validity of the findings

The analysis of the time complexity of the diffusion model is very vague. The comparison between diffusion and PLM is shallow and could barely provide any insights.

The review provides future directions in the field.

Cite this review as
Anonymous Reviewer (2024) Peer Review #1 of "Diffusion models in text generation: a survey (v0.1)". PeerJ Computer Science

Reviewer 2 ·

Basic reporting

The survey is written in very clear English and organized well. It is easy to read and navigate. The motivation, references and outlook to future research is very clearly stated.

The submission reviews a span of papers focusing on latest diffusion based based models used for NLP. The field has been reviewed recently but this submission focused on a narrow area, which is text generation. This area is an open research topic and this submission sheds lights and provides insights into future research.

Experimental design

This survey aims to give a thorough review of the latest developments in diffusion language models. It enhances readers understanding of this area. It categorizes research studies based on text generation tasks and offers in-depth explanations of the methodologies used. Additionally, it contrasts diffusion language models with pre-trained language models from various angles. The submission also discusses current challenges and potential future research avenues, offering valuable insights in related fields.

Validity of the findings

Please see additional comments. I think the paper could use some emprirical analysis reported from different papers.

Additional comments

Diffusion models has gained a lot of interest in the last few years and there is not good resource which significantly studies diffusion models for text generation. I believe this resource will be beneficial for the academic community and believe it will be cited.

This survey paper clearly summarize existing knowledge, points out how the field has progressed, and shows where more research could help. Instead of just listing existing studies, it compares different methods, spotlights key research, and hints at new trends. In short, the survey is an helpful tool that deepens understanding of the diffusion models for text generation.

Minor comments:
-- Not sure Figure 6 is necessary, autoregressive models are very well known. Fig 5 is enough to contrast against diffusion process in figure 7.
-- The submission misses a major part which compares the autoregressive and diffusion based models on some open source datasets. I would like to see a table where some common metrics are used as mentioned in "Diversity of generated results" section. This table should benchmark some known methods on the same datasets. This way we can see some performance difference. Another benchmark could be reported in another table perhaps, which should focus on the inference complexity and time. It would be beneficial to see how these models compare in terms of inference time complexity.

Cite this review as
Anonymous Reviewer (2024) Peer Review #2 of "Diffusion models in text generation: a survey (v0.1)". PeerJ Computer Science

---

## Round 0.2 · Minor Revisions

As the previous two reviewers weren't available this time, I have obtained a review from a new reviewer to check your revisions. This reviewer and I are both supportive of the publication of the work subject to some minor revisions, which you can see in the reviewer's comments.

Please do address the suggested revisions, although I would advise considering the addition of the specific references listed in the review as optional, as you may consider adding those or other suitable references to cover the discussion of multimodal diffusion models.

Reviewer 3 ·

Basic reporting

The paper conducts a comprehensive survey on the application of diffusion models in text generation. It categorizes text generation into conditional, unconstrained, and multi-mode text generation, providing a detailed exploration of each. The survey also compares diffusion models with autoregressive-based pre-training models (PLMs), discussing their advantages and limitations, and suggesting the integration of PLMs into diffusion models as a promising research direction. Current challenges and potential future research avenues, such as enhancing sampling speed for scalability and exploring multi-modal text generation, are discussed.
This survey is good enough for publication. The presentation is very clear and concise. Literature references is sufficient. The structure of suvery is good, readers can easy to understand the development of diffusion models in text generation. Furthermore, this survey also points some potential future research directions. These research direction is novel and interesting. Overall, I strong recommand to accept this survey with some revisions.
(1) It will be better, if authors add a Table or picture to describe the development of text generation diffusion models along the time.
(2) Summary the diffusion models on continous domain (such as, LDM) or discrete domain.
(3) Multimodal diffusion models. I agree with that multimodal diffusion models is very interesting in the future. However, authors donot investigate enough multimodal diffusion models (only one citation paper). Authors should deeply investigate multimodal diffusion models, and give suitable ciations. For example, many researchers has done text-to-audio generation and unified multimodal diffusion. Please see the following ciations.

[1] Ma Y, Yang H, Wang W, et al. Unified multi-modal latent diffusion for joint subject and text conditional image generation[J]. arXiv preprint arXiv:2303.09319, 2023.
[2] Yang D, Yu J, Wang H, et al. Diffsound: Discrete diffusion model for text-to-sound generation[J]. IEEE/ACM Transactions on Audio, Speech, and Language Processing, 2023.
[3] Huang R, Huang J, Yang D, et al. Make-an-audio: Text-to-audio generation with prompt-enhanced diffusion models[J]. arXiv preprint arXiv:2301.12661, 2023.
[4] Huang Q, Park D S, Wang T, et al. Noise2music: Text-conditioned music generation with diffusion models[J]. arXiv preprint arXiv:2302.03917, 2023.

Experimental design

no comment

Validity of the findings

no comment

Additional comments

The article "Diffusion Models in Text Generation: A Survey" provides a comprehensive overview of diffusion models in natural language generation (NLG). It categorizes text generation into conditional, unconstrained, and multi-mode text generation, comparing diffusion models with autoregressive-based pre-training models. The paper highlights the advantages, limitations, and potential future research directions of diffusion models in text generation, including improving sampling speed and exploring multi-modal text generation. This survey aims to serve as a valuable reference for researchers and practitioners in this field.

Cite this review as
Anonymous Reviewer (2024) Peer Review #3 of "Diffusion models in text generation: a survey (v0.2)". PeerJ Computer Science

---

## Round 0.3 · accepted · Accept

Thank you for carefully addressing the minor revisions recommended in the last round of reviews. Following my review of this last round of revisions, I consider that the manuscript is ready for publication in its current form and hence I recommend acceptance at this stage.